# SPA: Enhancing 3D Multimodal LLMs with Mask-based Streamlining Preference Alignment

## Abstract

Integrating 3D features into Large Language Models (LLMs) is a rapidly evolving field, with models like 3D-LLM, Point-Bind LLM, and PointLLM making notable strides. PointLLM, pre-trained and fine-tuned on the Objaverse dataset, enhances understanding by optimizing the projector, boosting resource efficiency and consistency. However, we observed a persistent bottleneck: increasing the LLM backbone size doesn't consistently improve performance. Preliminary experiments showed that enhancing the 3D encoder or extending fine-tuning alone failed to resolve this. While post-training partially addressed the issue, it required two stages and additional text sample generation, making it inefficient. To overcome this, we propose **S**treamlining **P**reference **A**lignment (**SPA**), a post-training stage for MLLMs with 3D encoders. SPA leverages the 3D encoder's inductive bias through 3D-masking, ensuring robust output while preserving consistent differences. Unlike traditional post-training, SPA maximizes the encoder's spatial reasoning by increasing the probability gap between positive and negative logits. This approach eliminates redundant text generation, greatly enhancing resource efficiency and improving the overall alignment process. In addition, we identified evaluation issues in the existing benchmarks and conducted a re-benchmark, resulting in a more robust evaluation approach. The model combined with the SPA method as post-training stage successfully overcame the performance bottleneck and achieved better results across various evaluations on current scene-level and object-level benchmarks. Code is available at https://anonymous.4open.science/r/3dmllm-dap-5A50.

## 1 Introduction

3D understanding plays a pivotal role in enabling accurate scene interpretation and object recognition, which are essential for a wide range of applications in robotics, augmented reality, and autonomous driving. Most previous studies have focused on extracting effective representations from point clouds and 3D meshes to improve downstream task performance. Approaches like PointNet (Qi et al., 2017a), PointNet++ (Qi et al., 2017b), and PointBERT (Yu et al., 2022) have made significant strides in this area. However, with the growing success of Multimodal Large Language Models (MLLMs) (Li et al., 2023; Liu et al., 2024a; Chiang et al., 2023), researchers are now exploring how these models can be applied to 3D data understanding. This trend has given rise to MLLM models with 3D encoders (Guo et al., 2023b; Hong et al., 2023), which combine point cloud features with text embeddings to enhance multimodal feature alignment and improve 3D object recognition and description tasks. In particular, PointLLM (Xu et al., 2023) simplifies the complex projector module in the past and brings 3D understanding into a new stage based on large-scale stable pre-training and fine-tuning alignment.

Despite the promising advancements of PointLLM, our investigation reveals a significant issue: performance bottleneck, i.e., a larger LLM backbone did not readily improve performance. As depicted in Figure 1, the performance of the 13B model is notably worse than the 7B model across various benchmarks, including zero-shot classification on ModelNet40 and caption generation on Objectaverse-based tasks. This performance bottleneck highlights the challenge of achieving generalization in larger models. Upon further investigation shown in Figure 3, we discovered that the root

cause of this issue lies in the misalignment of 3D features and text embeddings, which hampers the model's ability to effectively leverage its increased capacity.

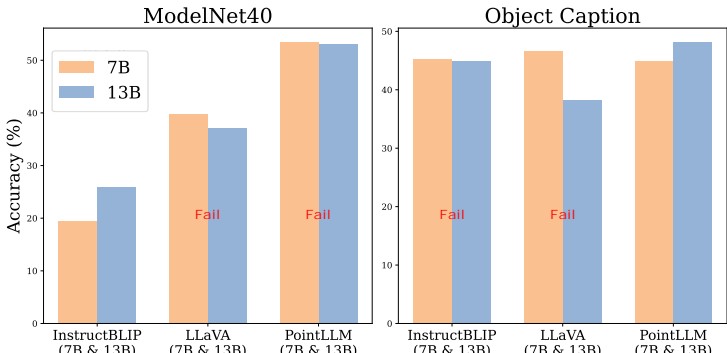

Figure 1: The performance bottleneck on different benchmarks shows that the performance of the 13B model is less than that of the 7B model on different benchmarks and different models. The left side represents the Generative 3D Object Classification tasks, and the right side represents the 3D Object Captioning tasks, which examines the generalization and language capabilities of the model.

A straightforward solution to address this misalignment is to apply Supervised Fine-Tuning (SFT). In SFT, the model is fine-tuned using multi-conversations 3D-text alignment data, which helps improve the accuracy of multimodal tasks. However, the drawback of SFT is its reliance on large-scale annotated datasets, which are expensive and time-consuming to obtain. To this end, we propose a novel approach: **S**treamlining **P**reference **A**lignment (**SPA**). Unlike traditional two-stage post-training methods, SPA simplifies the process by employing a one-stage alignment that uses ground truth as an anchor to guide the model's training. This reduces the complexity of fine-tuning and alleviates the limitations of SFT. The success of SPA stems from its ability to leverage 3D inductive biases through effective data augmentation strategies. Also, SPA ensures that the model can better capture the underlying spatial relationships between objects, leading to improved generalization across different tasks. This approach also allows for plug-and-play improvements in various downstream applications without the need for extensive retraining.

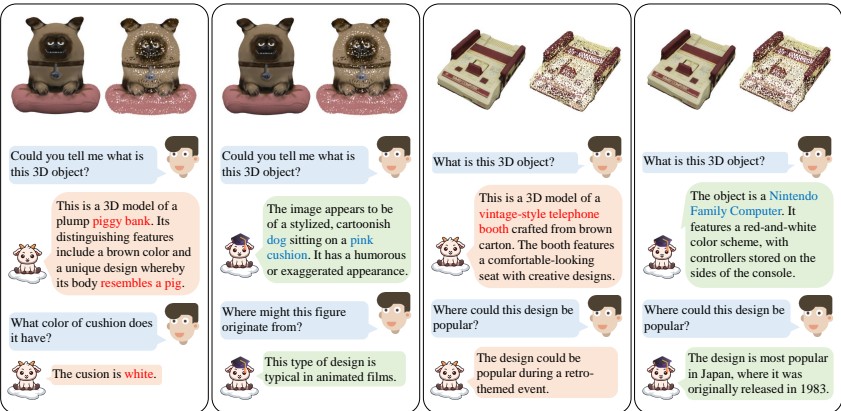

Figure 2: SPA provides improved answers compared to base model. The left image (brown) shows the conversation with PointLLM, while the right image (green) is model training with SPA.

To further validate the effectiveness of SPA, we repurposed existing datasets to create a comprehensive benchmark for evaluating 3D-MLLMs across multiple dimensions. Through extensive experiments, we demonstrate that SPA significantly improves performance on both object-level and scene-level tasks, surpassing existing methods in terms of accuracy and efficiency. As shown in some examples in Figure 2, after adding our method, the question answering problem in which PointLLM has errors becomes correct. Totally, our contributions are three-fold:

- We identify and investigate the performance bottleneck in current MLLMs with 3D encoders, providing empirical insights into their limitations.
- We introduce SPA, a novel post-training method that addresses misalignment issues and achieves optimal performance across several benchmarks.

- We re-benchmark existing evaluation frameworks to establish a more robust assessment methodology, facilitating a deeper understanding of 3D-MLLMs' capabilities.

## 2 PRELIMINARIES

**Injecting 3D encoders into LLM.** From 3D-LLM Hong et al. (2023) to Point-Bind LLM (Guo et al., 2023b), the integration of 3D modeling and MLLMs is advancing to a new stage. Notably, the success of PointLLM (Xu et al., 2023), built on large-scale pre-training and fine-tuning with Objaverse (Deitke et al., 2023; 2024), has marked a significant leap in 3D conversational capabilities. This approach offers substantial advantages over traditional 3D LLMs: it eliminates the need for cross-attention mechanisms like those in Q-former (Li et al., 2023), reduces training resource consumption, and enhances alignment capabilities.

**Preference Modeling in MLLMs.** In RLHF, the reward model was initially trained on preference pairs (Schulman et al., 2017). The training used a cross-entropy loss, treating binary choices—preference or rejection—as classification labels. This approach, known as the PPO strategy, maximizes the following objective:

$$\max_{\pi_\theta} \mathbb{E}_{x \sim D, y \sim \pi_\theta(y|x)} \left[ r_\phi(x, y) - \beta D_{\text{KL}}(\pi_\theta(y|x) \| \pi_{\text{ref}}(y|x)) \right], \tag{1}$$

where $x \sim D$ is the input, $y \sim \pi_\theta(y|x)$ is the output generated by the policy $\pi_\theta$, $r_\phi(x, y)$ is the reward model, $\beta$ is a scaling factor, and $D_{\text{KL}}(\cdot \| \cdot)$ is the Kullback-Leibler divergence between the learned policy $\pi_\theta$ and a reference policy $\pi_{\text{ref}}$. In the DPO (Rafailov et al., 2024) approach, the objective is further refined to:

$$L_{\text{DPO}}(\pi_\theta; \pi_{\text{ref}}) = \mathbb{E}_{(x, y^+, y^-) \sim D} \left[ -\log \sigma \left( \beta \log \frac{\pi_\theta(y^+|x)\pi_{\text{ref}}(y^-|x)}{\pi_{\text{ref}}(y^+|x)\pi_\theta(y^-|x)} \right) \right], \tag{2}$$

where $(x, y^+, y^-)$ are preference triplets, with $y^+$ as the preferred output and $y^-$ as the less preferred one, and $\sigma(\cdot)$ is the sigmoid function. In this context, the reward model is defined as a preference selection mechanism based on the Bradley-Terry (BT) theorem, which implicitly expresses preferences through acceptance or rejection. However, an additional step is required to generate outputs from the reference model $\pi_{\text{ref}}$ and ensure alignment with the learned policy $\pi_\theta$.

## 3 STREAMLING ALIGNMENT PREFERENCE MODELING

In this section, we begin by addressing the issue of inadequate model alignment in existing approaches. We then develop our method guided by empirical experiments. Through derivation, we demonstrate that our loss function is fundamentally equivalent to Information Noise-Contrastive Estimation (InfoNCE) (Oord et al., 2018), which indirectly elucidates the underlying mechanism of alignment insufficiency. This principle will be further analyzed in detail in the subsequent section 4.

### 3.1 UNDERALIGNMENT IN CURRENT METHOD

We have identified alignment deficiencies in existing methods (Xu et al., 2023; Hong et al., 2023), where significant scale effect anomalies are observed across different benchmarks in Figure 3. Figure 3(a) refers performance bottleneck with changing 3D encoder abilities, Figure 3(b) refers fine-tuning inefficiency and success in post-training stage. These anomalies likely stem from inadequate visual representation capabilities or misalignment issues. In our preliminary experiments, we found that model performance does not correlate directly with

encoder capacity Figure 3(a), revealing that the scale effect persists even when switching encoders including PointNeXt (Qian et al., 2022), PointNet2 (SSG) ](Qi et al., 2017b), PointMLP (Ma et al., 2022), PointBERT (Xue et al., 2023), PointBERT-ULIP2 (Xue et al., 2024b). Also, we find increasing the number of training epochs shown in Figure 3(b) can partially mitigate these anomalies, supervised fine-tuning extra 1 epoch (SFT 1ep) will lower original gap, supervised fine-tuning extra 2 epoch (SFT 2ep) can go futher. This strongly suggests that

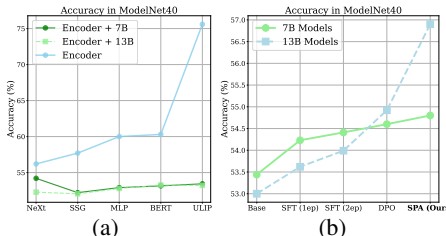

Figure 3: Empirical studies in PointLLM

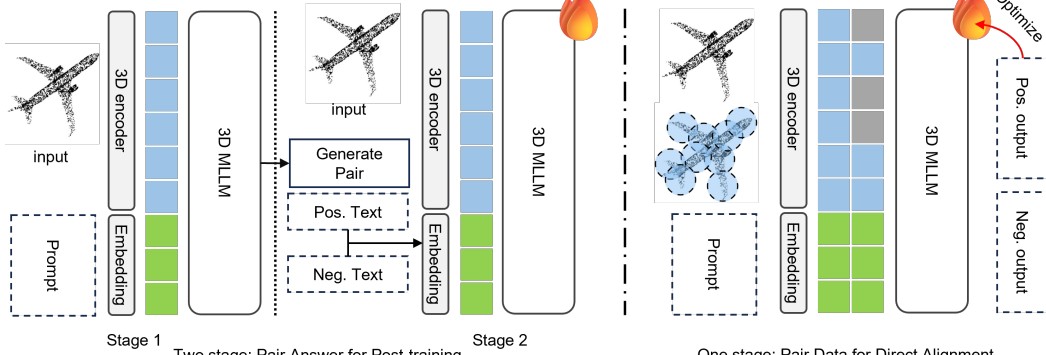

Figure 4: Standard post-training optimization involves aligning models with human preferences using reinforcement learning or reward models. In contrast, SPA generates preference-aligned data via symmetric noise sample inputs and directly optimizes the LLM based on differences in probability space, reducing the dependence on extra textual data generation. The **left figure** shows the typical post-training process, where paired data must be generated or obtained beforehand. The **right one** illustrates our framework, which enables Streamlining training and optimization directly on SFT data.

the core issue is related to alignment. But former method is both inefficient and resource-intensive. However, previous studies have shown that post-training techniques, particularly those utilized in MLLMs and LLMs, play a significant role in optimizing alignment, thereby effectively addressing various challenges. Therefore, we focus on post-training as a crucial supplementary phase in our alignment process. Drawing inspiration from established methods such as PPO (Schulman et al., 2017) and DPO (Rafailov et al., 2024), we propose **Streamlining Preference Alignment (SPA)**. This innovative approach is specifically designed to integrate point cloud features with LLMs, facilitating a more efficient solution of alignment issues while enhancing overall model performance.

### 3.2 STREAMLINING PREFERENCE ALIGNMENT MODELING FOR MLLMS WITH 3D ENCODER.

How to define a simpler post-training method that is suitable for 3D features? The key step lies in constructing preferred data pairs. Building on the foundation of previous self-supervised methods, we generate these preferred pairs by applying negative data augmentation to the input 3D data:

$$P(y|x_i) = \text{softmax}(f_{\text{LM}}(x_i)), \quad P(y|x_i') = \text{softmax}(f_{\text{LM}}(x_i')) \tag{3}$$

where $P(.)$ represents the probabilistic distribution in the space after the 3D data pair are encoded into features, processed through a projector, combined with textual embeddings, and passed through a LLM for predicting the next word. With such paired data $x_i$ and it's augmented negative input $x_i'$ we aim to maximize the divergence between the two probability distributions. Following the general post training framework, we derive the training objective as follows:

$$\mathbb{E}_{(x_i, x_i')} \left[ \log P(0|y) \right] = \mathbb{E}_{(x_i, x_i')} \left[ \log \sigma \left( \log P(y|x_i) - \log P(y|x_i') \right) \right] \tag{4}$$

where $\sigma(z) = 1/(1 + \exp(-z))$ is the sigmoid function which employed to transform the log-probability difference into a probability ranging between 0 and 1. The preference probability $P(v_i \succ v_i'|y)$ is derived using BT theorem to model pairwise ranking relationships. Notably, as the logits are generated dynamically based on multi-round conversational inputs, there is no need for additional paired data generated via a reference model. Returning to the loss formulation based on equation Figure 4, we express the loss as:

$$\mathcal{L} = -\log P(x_i \succ x_i'|y) = -\log \sigma \left( \log P(y|x_i) - \log P(y|x_i') \right) \tag{5}$$

Expanding and simplifying the expression yields:

$$\mathcal{L} = -\log \left( 1 + \exp \log \left( \frac{P(y|x_i')}{P(y|x_i)} \right) \right) = \log \frac{P(y|x_i)}{P(y|x_i) + P(y|x_i')} \tag{6}$$

At this stage, the reference model becomes unnecessary because our model alignment direction comes from the ground truth conversations itself rather than the output reference of the reference

model. As a result, our method effectively fine-tunes model outputs by conditioning them on 3D feature representations, which enables implicit preference modeling. This enhances the model's ability to distinguish between positive and negative samples, refining the decision boundary to better match the training objectives of InfoNCE. By directly optimizing the alignment direction, our approach integrates preference alignment with contrastive learning, eliminating the need for explicitly generating paired text data. This unified approach not only simplifies the learning process but also improves model efficiency by focusing on 3D features during contrastive training.

**Single effective stage framework**. In this work, we address the limitations of traditional two-stage post-training framework for MLLMs with 3D encoders, such as PointLLM, where the second stage typically neglects visual features, leading to suboptimal utilization of multimodal data and convergence to suboptimal outcomes. The two-stage post-training framework are defined as follows: in the first stage, preference data pairs are generated and a reference model is trained; in the second stage, preference learning, such as DPO, is performed based on the data generated in the first stage. Shown in Figure 4, in traditional post-training method, the first stage involves generating a set of preference texts, either by directly corrupting the ground truth or by corrupting the input prompts or 3D features to generate preference texts that are fed back into the model to compute probabilities across samples. Our approach simplifies the process by merging these two stages into a single-stage preference alignment, where visual representations are leveraged as priors to optimize the language probability space. By utilizing improved positive ground truths as anchor samples, our method enables tighter clustering of similar samples within the representation space, enhancing robustness against irrelevant

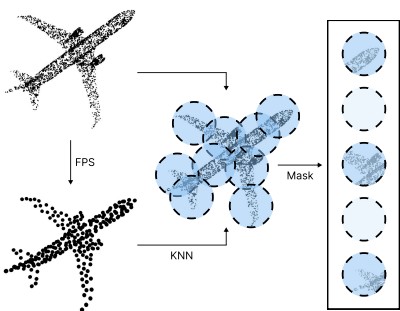

Figure 5: **3D Masking** method in point cloud input. We utilize FPS to select central points, followed by KNN to compute neighboring points. The point cloud is then partitioned into multiple circular regions, after which random masking is applied to these regions.

features. Unlike conventional methods that halt image utilization in the later stages and manipulate preference data for optimization, our framework ensures stable and efficient training by fully integrating the 3D encoder's visual representations throughout the process. This not only maximizes data utilization but also achieves superior alignment between the output logits and the positive ground truth, leading to significant performance improvements.

**Robust negative data augmentation mode**. We follow the approach proposed in (Guo et al., 2023a) and adopt 3D random masking as our data augmentation strategy which shown in Figure. This method helps stabilize the variability in output responses while ensuring that the generated outputs remain aligned with the inherent LLM-based QA framework. Compared to conventional data augmentation techniques, 3D random masking not only introduces diverse data patterns but also prevents the model from overfitting to specific input configurations, resulting in better generalization in generated answers. Furthermore, this approach strikes an effective balance between training complexity and model stability. A more detailed discussion of this trade-off, including its effects across different scenarios, is provided in the ablation studies presented in Section 4.

**Boost post-training starting from the supervised anchor.** The proposed SPA method effectively mitigates the limitations of previous post-training techniques in integrating 3D features with MLLMs. Notably, the anchors in SPA are derived from supervised labels, which, despite being less random than those used in DPO as reference models, provide a more stable and well-defined foundation for training. This strategic shift allows for a performance ceiling in the 3D domain that is less reliant on the model and data, and instead places greater emphasis on the data itself. As a result, this transition revitalizes the potential of self-supervised scaling laws, thereby enhancing the overall efficacy of our approach. Furthermore, as illustrated in Figure 6, we demonstrate that fine-tuning, which employs standard response outputs and labels to compute cross-entropy loss for boundary construction, can achieve a certain degree of discrimination. However, in the absence of negative samples, its generalization capability is constrained, complicating the handling of out-of-distribution scenarios. In contrast, post-training methods leverage positive samples as expectations to approximate anchors and increase the separation from negative samples, albeit introducing some error. Our SPA method synergistically combines the strengths of both approaches: it establishes

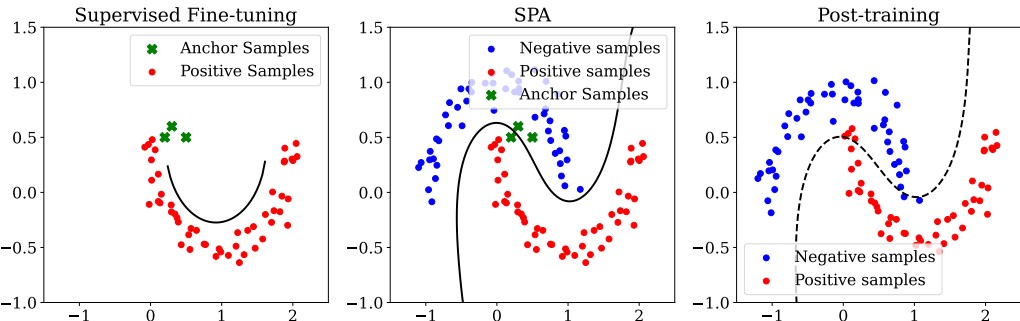

Figure 6: The influence of different learning modes on the decision boundary of the model is used. The anchor sample represents the label and its corresponding data, the positive sample represents the sample corresponding to the logit generated by the model for normal images, and the negative sample represents the sample corresponding to the logit generated by the model for noisy images. We obtain the logits output of the last few layers of LLM and convert them into corresponding probability distributions, and select two of the more critical feature dimensions for t-SNE dimensionality reduction drawing. The basic model is PointLLM, which uses data augmentation to generate positive and negative sample outputs, and uses ground truth as input to obtain the corresponding probabilities for drawing.

stable boundaries using labeled data while simultaneously enhancing the distance from negative samples to improve generalization performance, thereby achieving superior results.

## 3.3 REBENCHMARKING BENCHMARKS

In previous evaluation methods, traditional metrics like BLEU-1, ROUGE-L, and METEOR tend to favor shorter responses and may not effectively capture semantic accuracy. When using GPT-4 for evaluation, the direct comparison between the answer and ground truth text can lead to inaccuracies, often overlooking key factors in the question. Manual scoring, where human raters assign quantitative scores, may introduce variability and subjectivity across different evaluators. To address the challenges of instability and inconsistency often observed in existing GPT-level and human-level evaluations of benchmarks (Azuma et al., 2022; Vishwanath et al., 2009; Brazil et al., 2023; Luo et al., 2024), we propose a novel approach that leverages automated re-annotation based on pre-trained LLMs. By transforming descriptive annotations into structured, multi-choice question-answer formats, we introduce the **3D C**hoice-level **Q**uestions and **A**nswering (**3DCQA**) benchmark. This approach enables a more comprehensive evaluation at both object and scene levels, promoting a more reliable and interpretable framework for performance assessment. The benchmark introduces a structured question template based on selective questions of different capabilities to evaluate a range of 3D-related capabilities of MLLMs.

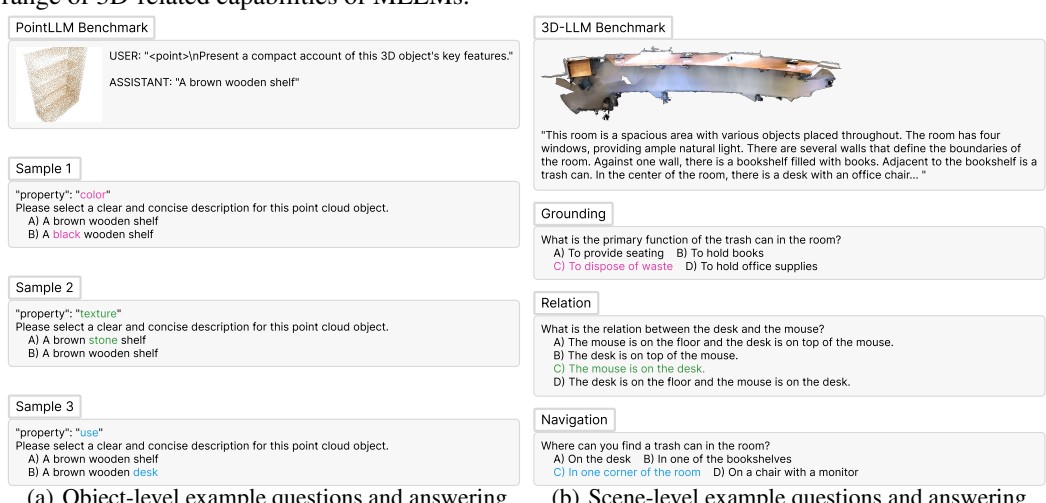

(a) Object-level example questions and answering      (b) Scene-level example questions and answering

Figure 7: **3DCQA Benchmark**.We repurpose standard 3D benchmarks to evaluate both object-level and scene-level abilities for MLLMs with 3D encoders.

Our benchmark comprehensively evaluate the models' understanding capabilities in performing both object recognition and internal scene spatial analysis. We draw on the 3D Object Captioning benchmark proposed by PointLLM (Xu et al., 2023) and 3D captioning ScanQA benchmark proposed by 3D-LLM (Hong et al., 2023) as our foundational material. For each data record, We utilize Llama-3.1 model to automatically generate multiple-choice questions for every category based on the respective data caption, and then let the language model select the question which can be reasonably inferred from the original caption as our benchmark.

At the **object** level shown in Figure 7(a), evaluations focus on fundamental object characteristics. This includes aspects such as color, texture, and functionality, which represent core features crucial for object recognition. Meanwhile, at the **scene** level shown in Figure 7(b), the framework delves into more advanced spatial and relational reasoning tasks. This includes object localization, where the model must identify not only the presence of objects but also their precise positions within a scene. It also encompasses navigation and the interpretation of spatial relationships, requiring the model to understand how objects relate to one another within the 3D space. These evaluations push the model to perform in scenarios that mimic real-world environments, testing its ability to make sense of complex spatial arrangements and navigate through dynamic and structured spaces. Together, these dimensions provide a comprehensive assessment of the model's capacity to interpret and engage with 3D environments, mirroring the intricacies encountered in real-world applications.

We selected a subset of samples and captions for rebenchmarking using the ScanQA test set and PointLLM's Objaverse caption benchmark. The table illustrates the number of samples associated with each ability category, capturing a wide range of competencies. By integrating these structured assessments into a unified framework, 3DCQA becnhmark provides a systematic and scalable approach for evaluating 3D understanding across various dimensions. This significantly reduces subjectivity and enhances consistency, ensuring that evaluations are objective and replicable. Moreover, the 3DCQA framework facilitates future research by offering a clear, structured methodology for identifying gaps in model performance, thus pinpointing areas for potential improvement. Its abstracted yet robust evaluation design enables broader applicability, covering different types of 3D models and a wide variety of tasks. This approach not only drives more comprehensive assessments but also challenges current methodologies, pushing the boundaries of 3D model capabilities and encouraging ongoing innovation in the field. More details show in the B.

## 4 EXPERIMENTS

### 4.1 EXPERIMENTAL SETUP

**Implementation Details.** Following PointLLM (Xu et al., 2023), we employ the LLaMA architecture as the foundation LLMs, specifically utilizing checkpoints from the 7B and 13B variants of Vicuna (Chiang et al., 2023). For the encoding of point clouds, we adopt Point-BERT (Yu et al., 2022), pretrained on the Objaverse dataset (Deitke et al., 2023) via ULIP-2 (Xue et al., 2024a). Notably, the 200 objects from Objaverse utilized in our benchmarks are excluded from all training phases to ensure impartial evaluations. Each point cloud is represented by $n = 8192$ points, each comprising $d = 6$ dimensions. In the absence of color information for datasets like ModelNet40, we uniformly assign a black color to the point clouds. The point encoder generates $m = 513$ features, each with a dimensionality of $c = 384$. These features are subsequently processed through a projection module, consisting of three linear layers with GeLU activation (Hendrycks & Gimpel, 2016), mapping them to tokens of dimension $c' = 5120$ for both the 7B and 13B models. We also introduce two special tokens, resulting in a total vocabulary size of $V = 32003$ for PointLLM. All experiments are conducted on a distributed setup of $4 \times 80GB$ NVIDIA A100 GPUs. The GPT-4 and ChatGPT models referenced herein align with OpenAI's "gpt-4-0613" and "gpt-3.5-turbo-0613", respectively.

### 4.2 COMPARISON RESULTS FOR DIFFERENT ABILITIES

As shown in Table 1, through comparative experiments on a general benchmark, we evaluate the classification and captioning capabilities of the model itself, where the former is evaluated by classification task usining prompt **"What is this?"**, and the latter is evaluated by GPT-4 and prompted for shorter captions with no more than 20 words. Notably, SPA significantly addresses the critical issue of LLM backbones with less than 7B parameters, which has persisted in prior research. Our

Table 1: Generative 3D object results on the **General** and **Choice** Benchmark. General Benchmark includes two tasks **Generative 3D Object Classification** and **3D Object Captioning**. We select ModelNet40 **(M40.)** test split and Objaverse Caption **(Obj.Cap.)** as representative subset. Choice Benchmark is introduced in Sec. 3.3.

| Model | Input | General | | Choice | |
|---|---|---|---|---|---|
| | | M40. | Obj.Cap. | Sce.QA (c) | Obj.QA (c) |
| InstructBLIP-7B | Sin.-V. Img. | 19.53 | 45.34 | 27.11 | 44.21 |
| InstructBLIP-13B | Sin.-V. Img. | 25.97 | 44.97 | 29.23 | 39.17 |
| LLaVA-7B | Sin.-V. Img. | 39.75 | 46.71 | 33.21 | 66.17 |
| +SPA | Sin.-V. Img. | 41.11 | 45.92 | 35.75 | 65.23 |
| LLaVA-13B | Sin.-V. Img. | 37.12 | 38.28 | 31.55 | 64.92 |
| +SPA | Sin.-V. Img. | 42.09 | 44.19 | 36.83 | 67.94 |
| 3D-LLM | 3D + Mul.-V. | - | 33.42 | 45.12 | - |
| PointLLM-7B | 3D Data | 53.44 | 44.85 | 11.30 | 73.33 |
| +SPA | 3D Data | 54.80 | 46.77 | 36.97 | 76.89 |
| PointLLM-13B | 3D Data | 53.00 | 48.15 | 12.19 | 70.59 |
| +SPA | 3D Data | **56.90** | **54.07** | **43.85** | **79.08** |
| Average Gain | | **+2.90** | **+3.24** | **+16.29** | **+3.53** |

Table 2: Replace the ablation experiment with different noise levels and different noise types to explore the impact of negative data augmentation on the results. Evaluation includes ModelNet40 and Objaverse Caption which is same as Table 1.

| Noise level | Obj.Cap. | M40. |
|---|---|---|
| mask 10% | 46.77 | 54.80 |
| mask 25% | 46.12 | 55.11 |
| mask 50% | 45.98 | 55.23 |
| mask 75% | 45.18 | 55.13 |

| Noise type | Obj.Cap. | M40. |
|---|---|---|
| Mask | 46.77 | 54.80 |
| Gassion | 44.77 | 53.98 |
| Random Drop | 45.11 | 54.12 |

model demonstrates substantial improvements even in single-view image scenarios, such as LLaVA, highlighting the significant impact of our approach on the model's spatial capabilities. Similarly, the results on the choice-related benchmark, 3DCQA, can be analyzed from a more detailed perspective. More results are shown in C.

Shown in Table 3, we conduct experiments on 13B PointLLM and follow setting same as Table 1 choice benchamrk part. In the scene-level experiments, it was observed that the 13B PointLLM model initially exhibited limited performance when engaging in scene-related conversations. This shortfall can be attributed to the model's pre-training process, which lacked sufficient exposure to rich, scene-specific datasets, and the absence of tailored fine-tuning. However, after undergoing additional rounds of supervised fine-tuning (SFT), the model demonstrated substantial improvements. Notably, in navigation-related tasks, the model's performance reached a satisfactory level, particularly due to the integration of scene-relevant knowledge during fine-tuning. This highlights the importance of domain-specific adaptation in enhancing model proficiency for specialized tasks. In contrast, the SPA method consistently outperformed PointLLM in scene-related tasks, particularly by effectively improving the model's grounding and relational reasoning capabilities. This can be attributed to SPA's ability to establish more robust decision boundaries, especially for judgment-based problems. These clear demarcations enable the model to better handle complex relational queries, offering a significant advantage in tasks that require spatial reasoning or contextual understanding. On the object-level, the initial performance of the 13B PointLLM was commendable in conversations that revolved around object-specific queries, such as identifying attributes like color or texture. However, a surprising trend emerged extra fine-tuning: the model's generalization ability declined, particularly in tasks involving subtle distinctions in color and texture selection. This regression in performance highlights a potential overfitting issue, where the model becomes too specialized to the fine-tuning dataset, losing its adaptability to broader queries. In contrast, the SPA method exhibited a remarkable ability to mitigate these challenges. Even when trained under large-scale pre-training conditions, SPA maintained stable performance gains, effectively preserving its generalization capability across object-related tasks.

Table 3: Detail results on 3DCQA benchmark, 13B PointLLM Compared to use additional supervised fine-tuning (SFT) 1 epoch and post-training by SPA.

| Scene level | | | |
|---|---|---|---|
| | Base | +SFT | +SPA |
| Grounding | 16.84 | 40.69 | **53.27** |
| Relation | 14.57 | 35.24 | **48.15** |
| Navigation | 6.40 | 29.55 | **38.48** |
| Object level | | | |
| | Base | +SFT | +SPA |
| Color | 84.62 | 74.36 | **87.17** |
| Texture | 76.19 | 69.05 | **80.95** |
| Use | 59.72 | 66.67 | **73.61** |

## 4.3 ANALYSIS AND ABLATION

**Exploration of data augmentation.** For negative data augmentation, we consider an analysis along two dimensions: noise level and noise type. The former may affect the construction of the model's decision boundary, while the latter may influence the shift in the as-

sociated probability distribution. As shown in Table 2, it is evident that the optimal noise level falls between 25% and 50%. Compared to random dropping and adding Gaussian noise, the 3D masking method demonstrates superior linguistic expression and generalization, likely due to the inherent characteristics of point cloud data and the properties of the 3D encoder. Since point cloud compression requires downsampling, the FPS in the 3D masking step precisely selects core points. By randomly masking these point cloud clusters, we effectively obscure areas critical to visual representation, resulting in stable differential outcomes.

**Post-training and preference modeling.** In Table 4, we provide a comprehensive comparison of existing two-stage post-training methods, specifically DPO (Rafailov et al., 2024) and SimPO (Meng et al., 2024). To thoroughly demonstrate the efficacy of our proposed approach, we implemented two distinct modes for generating text pairs. The first method involves directly employing the LLM to rewrite and generate negative text, a process we refer to as text corruption. This approach allows us to leverage the model's generative capabilities to create text that diverges from the desired output. The second method is a more sophisticated data augmentation technique that harnesses the internal knowledge of the model in conjunction with SPA, more details setting shown in A.2. In Stage 1 of this process, we introduce masking to the input point cloud to generate negative text, while the unmasked output serves as the positive text reference.

Table 4: Results on the General benchmark, following Table 1's settings. The Fine-tuning, DPO and SimPO methods are compared including data augmentation (DA) and text corruption (TC) to generate text pair.

| Model | Obj.Cap. | M40. |
|---|---|---|
| Base | 48.15 | 53.00 |
| +SFT | 48.88 | 53.62 |
| +SPA | **54.07** | **56.90** |
| +DPO (DA) | 50.01 | 54.92 |
| +SimPO (DA) | 49.95 | 54.12 |
| +DPO (TC) | 50.71 | 53.77 |

This dual approach not only enhances the diversity of the generated text pairs but also ensures that the model can learn from both the corrupted and valid instances. Our comparative analysis reveals that the DPO method exhibits superior generalization and performance, particularly in classification tasks, despite showing slightly diminished effectiveness in captioning tasks. In contrast, SimPO, as a streamlined version of DPO that operates without a reference model, mirrors this trend but falls short of DPO's performance metrics. These findings compellingly illustrate that the SPA method not only maintains robust performance across various tasks but also surpasses previous post-training methodologies, thereby establishing its superiority in enhancing model performance.

## 5 RELATED WORKS

Recent years have seen remarkable progress of MLLMs (Li et al., 2023; Liu et al., 2024a; Chiang et al., 2023), leveraging their outstanding zero/few-shot reasoning performance of LLMs on vision-language and other modality tasks (Brown, 2020; Chowdhery et al., 2023; Team, 2023; Touvron et al., 2023). Efforts to empower MLLMs to better comprehend information across these modalities have focused on MLLM key components including (i) MLLM Backbone, (ii) Visual Encoder, and (iii) Post-training Strategy. In this section, we investigate these key aspects related to 3D vision understanding and reasoning.

**Multimodal LLMs.** The integration of multiple modalities in MLLMs, particularly vision and text, has become increasingly prominent since GPT-4V revealed remarkable generalization capabilities on these modalities (Yang et al., 2023). Earlier studies also discovered the potential of language models to perform 3D comprehension in the 2D image modality (Brazil et al., 2023; Tong et al., 2024a). 3D-LLM (Hong et al., 2023) constructs representation of 3D scenes by extracting 2D feature from multi-view images and performs computationally inefficient cross-attention. Inspired by ImageBind (Girdhar et al., 2023), Point-Bind LLM (Hong et al., 2023) binds point cloud information with images for cross-modal retrieval and downstreaming tasks. Specifically, PointLLM (Xu et al., 2023) proposed an end-to-end point cloud alignment paradigm utilizing conventional 3D feature extractor PointBERT (Yu et al., 2022) which focus on capturing 3D geometric structures and effectively representing point clouds. Despite these advancements, issues such as poor generalization on unseen data and high computational costs in post-training phases have persisted, limiting further practical applications.

**3D Visual Encoder.** Typical MLLMs utilize language-supervised visual encoders such as CLIP (Radford et al., 2021) to exploit similarity and bridge visual-text modalities. This inspired PointCLIP (Zhang et al., 2022), PointCLIPv2 (Zhu et al., 2023) and CLIP2Point (Huang et al., 2023b), which transform point clouds to depth maps within this framework. In contrast, ULIP (Xue et al., 2023),

ULIP-2 (Xue et al., 2024b), CG3D (Hegde et al., 2023) and OpenShape (Liu et al., 2024b) follow the CLIP contrastive learning fashion to extract 3D features supervised by CLIP image-text embedding. Another branch of point cloud encoders follow PointNet (Qi et al., 2017a) and PointNet++ (Qi et al., 2017b) and build transformer models leveraging rotation invariance, including Point Transformer (Zhao et al., 2021), Point Cloud Transformer (PCT) (Guo et al., 2021) and PointBERT (Yu et al., 2022), ushering a concise end-to-end encoder design.

**Non-Streamlining Post-training Preference Alignment.** Preference alignment and optimization strategies have been widely studied and adopted in the domain of LLMs and MLLMs to mitigate hallucinations and ethical challenges of generating malicious content (Huang et al., 2023a; Jiao et al., 2024), paving the way for a wide range of alignment methodologies (Ouyang et al., 2022; Shen et al., 2023). In terms of optimization algorithms, RLHF approaches represented by PPO (Schulman et al., 2017) employ policy gradient methods to optimize a reward function, resulting in impressive performance but high computational costs and sample inefficiencies. To address this issue, DPO (Rafailov et al., 2024) proposes a direct optimization objective of policy model that is trained on candidate output pairs in an offline fashion. This progress has encouraged the emergence of more theoretically grounded modifications based on DPO. Identity-PO (Azar et al., 2024) uses identity mapping to directly optimize pairwise preferences and removes reliance on ELO scores to avoid overfitting problem in DPO. R-DPO (Park et al., 2024) introduces a length regularization term to overcome verbosity caused by over-exploitation of length. SimPO (Meng et al., 2024) eliminates the reference model with average log-likelihood as an implicit reward.

## 6    CONCLUSION AND FUTURE WORKS

**Conclusion.** In conclusion, 3D understanding remains a critical component in advancing technologies like robotics, augmented reality, and autonomous driving. While previous approaches have contributed significantly to enhancing 3D object recognition and description tasks, the integration of MLLMs with 3D encoders introduces new possibilities for aligning text and 3D features. Our study highlights the performance limitations encountered with larger model backbones, demonstrating that increased capacity does not necessarily translate to better performance due to feature misalignment. To address this, we proposed the SPA method, which simplifies the post-training process and improves model performance through one-stage fine-tuning. Our extensive experiments confirm the effectiveness of SPA in enhancing accuracy and generalization across a range of tasks. This work contributes valuable insights into 3D-MLLMs and lays the foundation for future research in multi-modal feature alignment and 3D data understanding.

**Future work.** Looking ahead, we propose several important directions for future research. First, developing more efficient model architectures is essential to reduce computational overhead and improve real-time performance, particularly for applications with limited resources. Second, further exploration of cross-modal alignment techniques, especially in dynamic and complex environments, could enhance model adaptability and accuracy. Finally, utilizing a broader range of diverse datasets will be key to strengthening model robustness and improving generalization across tasks and settings. Advancing these areas will deepen our understanding of 3D data processing and drive innovation across multiple fields, from robotics to immersive technologies.

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

# APPENDIX

The appendix is structured as follows:

(A) In Appendix A, we provide implementation details are provided including fine-tuning, pre-training and post-training settings.

(B) In Appendix B, we describe more details and provide examples in 3DCQA benchmark.

(C) In Appendix C, we provide additional experimental results as support.

(D) In Appendix D, we further provide extensive related work to highlight connections and differences to the proposed approach.

# A  IMPLEMENTATION DETAILS

## A.1  PRETRAIN AND FINE-TUNING

Following PointLLM, We also train PointLLM by minimizing the negative log-likelihood of text tokens at each position. The loss is computed only for the text tokens that constitute the model's responses, including the end-of-sentence token $$, while excluding tokens from human instructions. This strategy ensures that the model can focus on generating accurate and coherent outputs. Such an end-to-end training approach allows PointLLM to efficiently integrate point cloud and text information. Our training process is divided into two stages, each focusing on different aspects of the model. In the first stage, known as the pre-training stage, we freeze the parameters of the point cloud encoder and the LLM, training only the MLP projector. During this phase, we use brief descriptive instructions aimed at effectively aligning point features with the text token space and adjusting the embeddings for the newly added special tokens $<p\_start>$ and $<p\_end>$. In the second stage, referred to as the instruction tuning or fine-tuning stage, we freeze the point cloud encoder while jointly training the projector and the LLM. This stage employs complex instructions to enhance the model's ability to understand and respond to intricate commands, including those involving point cloud data.

## A.2  POST-TRAINING

Table A1: Preference optimization objectives and hyperparameter search range.

| Method | Objective | Hyperparameter |
|---|---|---|
| DPO (Rafailov et al., 2024) | $-\log\sigma\left(\beta\log\frac{\pi_\theta(y_w|x)}{\pi_{\text{ref}}(y_w|x)} - \beta\log\frac{\pi_\theta(y_l|x)}{\pi_{\text{ref}}(y_l|x)}\right)$ | $\beta \in [0.01, 0.05, 0.1]$ |
| SimPO (Meng et al., 2024) | $-\log\sigma\left(\frac{\beta}{|y_w|}\log\pi_\theta(y_w|x) - \frac{\beta}{|y_l|}\log\pi_\theta(y_l|x) - \gamma\right)$ | $\beta \in [2.0, 2.5]$ $\gamma \in [0.3, 0.5, 1.0, 1.2, 1.4, 1.6]$ |
| SPA | $-\log\sigma\left(\log\pi_\theta(y|x_w) - \log\pi_\theta(y|x_l)\right)$ | |

For post-training, we refer to the intrinsic and extrinsic corruption methods in the (Zhou et al., 2024) paper and directly modify the text or corrupt the image (same as SPA) to create text pairs. In text corruption, our goal is to generate unpleasant hallucination responses by hallucinating the real correct response. In this, we use GPT as an editing method to directly edit the current answer as part of the data. In text corruption, our goal is to generate unpleasant hallucination responses by hallucinating the real correct response. In this, we use GPT as an editing method to directly edit the current answer as part of the data. After that, for the DPO method, our $\beta$ is set to $0.1$, for the SimPO method, our $\beta$ is set to $2.0$, and $\gamma$ is set to $0.3$. This makes it reach a better level in the reference hyperparameter setting for fair comparison.

# B BENCHMARK DETAILS

Table A2 provides a detailed breakdown of 3DCQA, which includes scene-level and object-level question-answering tasks, including the scene level part based on the ScanQA test set and the objectaverse question-answering based on PointLLM. At the scene level, there are 827 questions categorized into grounding, relation, and navigation, with 300, 284, and 243 questions, respectively. At the object level, there are 153 questions divided into color, texture, and use, with 39, 42, and 72 questions, respectively. This distribution indicates a balanced approach between understanding complex 3D scenes and focusing on specific object attributes. We provide detailed examples in Figures A1 and A2.

Table A2: Detailed information of 3DCQA.

| Scene level | | | |
|---|---|---|---|
| Grounding | Relation | Navigation | Total |
| 300 | 284 | 243 | 827 |
| Object level | | | |
| Color | Texture | Use | Total |
| 39 | 42 | 72 | 153 |

| Objaverse id | Type | Question | Answer |
|---|---|---|---|
| 267b8ecaf288 4abaaa5f0368 0ce39ad8 | Color | *What is the color of the computer cpu? A) Silver. B) Black. C) Red. D) Blue.* | *B) Black.* |
| a9fa3b6a1da7 4b5aa369250 12f251638 | Color | *Please select a clear and concise description for this point cloud object. What is the color of the smartphone? A) The smartphone is black, but the screen saver is blue. B) The smartphone is blue, but the screen saver is black. C) The smartphone is black with a blue colored screen saver. D) The screen saver is blue, but the smartphone is black.* | *C) The smartphone is black with a blue colored screen saver.* |
| 0031ba19d3e 042c4bcf79eb a40ccc812 | Texture | *Please select a clear and concise description for this point cloud object. A) The object is smooth and shiny. B) The object has a rough and bumpy texture. C) The object has a rough and bumpy texture on the sides and a smooth and shiny texture on the legs. D) The object is a white container like car with six black tractor legs and yellow sides.* | *B) The object has a rough and bumpy texture.* |
| 245af7dde0cd 4add9f7e11db 3bbbccba | Texture | *Please select a clear and concise description for this point cloud object. A) Smooth and glossy, like polished metal. B) Rough and bumpy, like a rocky terrain. C) Yellow colored blue glassed submarine. D) Soft and fluffy, like a feather.* | *C) Yellow colored blue glassed submarine.* |
| 0ea33b66171 74530b97d6b 7a92c275fb | Use | *What is the object used for? A) A decorative centerpiece for a table. B) A toy for children to play with. C) A cartoon green and red like a fruit. D) A kitchen appliance for cooking food.* | *C) A cartoon green and red like a fruit.* |
| c20eb3a5a93e 4cddb06c2f98 626b1830 | Use | *Please select a clear and concise description for this point cloud object. A) As a decorative item in a living room. B) A wooden rectangular board with a clay pot on a three stand and a table having some utensils on top. C) A cooking utensil in a kitchen. D) A storage container in a garage.* | *B) A wooden rectangular board with a clay pot on a three stand and a table having some utensils on top.* |

Figure A1: Object-level examples from our 3DCQA benchmark. We categorize question types into color, texture, and use. Different question types vary on their testing focuses.

| ScanNet id | Type | Question | Answer |
|---|---|---|---|
| scene0264_00 | Grounding | *What is the location of the bulletin board in the room? A) On the floor. B) On one of the walls. C) On the desk. D) On the shelf. Please answer directly with only the letter of the correct option and nothing else.* | *B) On one of the walls.* |
| scene0399_00 | Grounding | *What is located above the two sinks in the bathroom? A) A single mirror. B) A toilet. C) Two mirrors. D) A paper towel dispenser.* | *C) Two mirrors.* |
| scene0079_00 | Relation | *What is the relationship between the copier and the printing or copying needs in the room? A) The copier is the source of the printing or copying needs. B) The copier is used to assist with the printing or copying needs. C) The copier is unrelated to the printing or copying needs. D) The copier is the destination of the printing or copying needs.* | *B) The copier is used to assist with the printing or copying needs.* |
| scene0484_00 | Relation | *What is the relation between the two couches in the room? A) They are perpendicular to each other. B) They are parallel to each other. C) They are at opposite corners of the room. D) They are at the same corner of the room.* | *B) They are parallel to each other.* |
| scene0022_00 | Navigation | *From the chair, which direction would you need to move to get to the bulletin board? A) Left. B) Right. C) Forward. D) Backward.* | *B) Right.* |
| scene0171_00 | Navigation | *Which part of the room allows natural light to enter and provides a view of the outside? A) The door on one of the walls. B) The window on one of the walls. C) The bookshelf. D) The floor.* | *B) The window on one of the walls.* |

Figure A2: Scene-level examples from our 3DCQA benchmark. We categorize question types into grounding, relation and navigation.

## C  EXTENSIVE EXPERIMENTS

Table A3: Generative 3D object results on two tasks **Generative 3D Object Classification** and **3D Object Captioning**. We select ModelNet40 **(M40.)** test split and Objaverse Caption **(Obj.Cap.)** as representative subset.

| Model | Input | Classification | | | | Caption | |
|---|---|---|---|---|---|---|---|
| | | M40.(I) | M40.(C) | Obj.(I) | Obj.(C) | GPT-4 | Sen.-BERT |
| InstructBLIP-7B | Sin.-V. Img. | 19.53 | 31.48 | 45.00 | 42.00 | 45.34 | 47.41 |
| InstructBLIP-13B | Sin.-V. Img. | 25.97 | 31.40 | 37.00 | 31.50 | 44.97 | 45.90 |
| LLaVA-7B | Sin.-V. Img. | 39.75 | 39.67 | 49.50 | 50.50 | 46.71 | 45.61 |
| +SPA | Sin.-V. Img. | 41.11 | 40.00 | 50.00 | 51.50 | 45.92 | 46.11 |
| LLaVA-13B | Sin.-V. Img. | 37.12 | 36.06 | 53.00 | 50.50 | 38.28 | 46.37 |
| +SPA | Sin.-V. Img. | 42.09 | 39.75 | 53.50 | 51.50 | 44.19 | 46.90 |
| 3D-LLM | 3D + Mul.-V. | - | - | 49.00 | 41.50 | 33.42 | 44.48 |
| PointLLM-7B | 3D Data | 53.44 | 51.82 | 55.00 | 51.00 | 44.85 | 47.47 |
| +SPA | 3D Data | 54.80 | 53.00 | 54.50 | 52.00 | 46.77 | 47.37 |
| PointLLM-13B | 3D Data | 53.00 | 52.55 | 56.50 | 51.50 | 48.15 | 47.91 |
| +SPA | 3D Data | **56.90** | **55.33** | **57.00** | **52.50** | **54.07** | 46.61 |

Table A3 shows more experiment results. The results in generative 3D object classification show the classification accuracy under the instructive (I) cue "What is this?" and the completion (C) cue "This is an object" as well as the average accuracy. For object caption, evaluation encompassesGPT-4 assessments and supplemented by Sentence-BERT which tend to favor shorter responses and may not effectively capture semantic accuracy and detailed discussion on (Xu et al., 2023). It is not difficult to observe that in the Open-vocabulary classification, as shown in the Table, our method essentially performs as an in-distribution classification, which corresponds to the distribution of the same set of 3D features. The open-vocabulary capability typically originates from the LLM, so with no changes made to the LLM itself, the performance improvement achieved by our method is marginal.

## D  EXTENSIVE RELATED WORKS

### D.1  ENHANCE MLLMs WITH VISION ENCODER

Recent achievements of multi-modal large language models (MLLMs) can be viewed as efforts to transfer the remarkable emergent capabilities demonstrated by large language models (LLMs) in natural language processing to the domain of computer vision. While large vision models (LVMs) excel at visual understanding and task-specific performance (Kirillov et al., 2023; Dosovitskiy, 2020; Nichol et al., 2021), they generally lack the broader reasoning abilities characteristic of LLMs (Yin et al., 2023). A pioneering contribution in this area is **LLaVA** (Liu et al., 2024a), which connects multimodal projector CLIP (Radford et al., 2021) with the pre-trained LLM Vicuna to create a visually aligned instruction-following model. Despite its simplicity, LLaVA effectively demonstrates how transformer modules can capture visual semantics and use them for downstream tasks. Along similar lines, **BLIP-2** (Li et al., 2023) introduces the Query Transformer (Q-Former) architecture to learn query-based visual semantics, eliminating the need for a full cross-attention mechanism and improving computational efficiency.

Other notable approaches further enhance these capabilities. **PaLI-X** (Chen et al., 2023) integrates a shared multi-modal transformer architecture to handle a variety of tasks including image captioning and visual question answering, while **Flamingo** (Alayrac et al., 2022) uses a lightweight gated cross-attention mechanism to fuse image and text representations, allowing models to perform zero-shot tasks across modalities with greater fluidity. These models extend the boundaries of what MLLMs can achieve by blending visual and textual data in more efficient and scalable ways.

While language-supervised MLLMs like LLaVA and BLIP-2 have demonstrated impressive performance, other research, such as **DINO** (Caron et al., 2021) and **DINOv2** (Oquab et al., 2023), focuses on self-supervised visual semantic extraction. These models aim to learn visual representations without explicit language supervision, enhancing model robustness in challenging visual tasks such as visual question answering (VQA). Empirical evidence suggests that self-supervised models,

such as DINO and DINOv2, can lead to more robust performance in tasks requiring visual reasoning and understanding, especially in real-world settings (Tong et al., 2024b).

To further evaluate these advancements, we designed the **3DCQA benchmark** and rebenchmarking process, which are specifically tailored to assess visual reasoning and understanding in complex, real-world environments. By focusing on a diverse range of scenarios, the benchmark provides a rigorous test of MLLM capabilities in the wild, enabling more comprehensive evaluations of how well these models generalize across tasks and modalities. This new benchmark is expected to push the field forward by setting a higher standard for visual understanding and reasoning.

### D.2 INJECTING 3D INTO MLLMS

The success of MLLMs on 2D images has inspired research to expand their capabilities to 3D modalities, aiming to capture richer geometric information and spatial context. This expansion into the 3D domain can be categorized into two main tasks: (i) **Object-level** tasks, which focus on recognizing and understanding individual objects in 3D space, and (ii) **Scene-level** tasks, which involve understanding the spatial relationships, layout, and navigation within complex scenes. To tackle these tasks, researchers have developed two predominant approaches for constructing 3D representations: (i) encoding point clouds directly from 3D data, and (ii) generating and processing multi-view images of 3D objects or scenes. Both approaches aim to leverage MLLM capabilities to interpret 3D data, but they differ significantly in their methodologies.

Point cloud encoders directly process 3D point cloud data to extract geometric features, which can then be aligned with textual and visual information. For example, LL3DA (Chen et al., 2024) employs a scene-level point cloud encoder to align 3D visual prompts with textual instructions, enabling the model to perform tasks such as navigation and interaction within 3D spaces. This approach allows the model to learn directly from raw 3D data, capturing detailed geometric features. Similarly, Point-Bind LLM (Guo et al., 2023b), inspired by ImageBind (Girdhar et al., 2023), aligns 3D object point clouds with multiple modalities, including images, text, and even audio. By doing so, it bridges the gap between 3D object recognition and multi-modal understanding. PointLLM, on the other hand, leverages PointBERT (Yu et al., 2022) as its point cloud encoder, capitalizing on the inductive biases inherent in 3D objects, such as symmetry and surface geometry. This allows the model to effectively process and understand 3D structures at an object level.

In contrast to point cloud encoders, another line of research focuses on generating multi-view images from 3D objects and scenes. These methods create 2D projections from different angles and then extract features using 2D-based models, such as CLIP (Radford et al., 2021). For instance, 3D-LLM (Hong et al., 2023) and Scene-LLM (Fu et al., 2024) render multiple 2D views from 3D data and use pre-trained image-text models to construct 3D representations. By projecting 3D objects into 2D space, these methods can leverage the strong prior knowledge embedded in 2D models, making them highly effective for tasks like scene understanding and object recognition in 3D contexts. One of the recent advancements in this area is LLAVA-3D (Zhu et al., 2024), which integrates multi-view image rendering with additional 3D information such as depth, camera position, and other spatial observations. By learning 3D positional embeddings, LLAVA-3D combines the strengths of 2D image-text alignment models with 3D spatial reasoning, resulting in a framework that can interpret complex 3D scenes. This approach effectively leverages the pre-existing 2D priors learned from MLLMs, while incorporating crucial 3D positional information, making it one of the most comprehensive frameworks for 3D representation learning.

The distinction between point cloud encoders and multi-view image-based methods highlights different strengths and limitations. Point cloud encoders offer direct access to 3D geometric information, making them ideal for fine-grained object-level recognition and manipulation. However, they often require specialized architectures to handle sparse and unordered data. In contrast, multi-view image-based approaches benefit from the well-established success of 2D models but may struggle to fully capture the depth and geometric nuances of 3D data, as they rely on 2D projections. Future research will likely continue to explore ways to combine the strengths of both approaches. For example, integrating point cloud encoding with multi-view rendering could provide richer representations by fusing raw 3D data with the powerful priors learned from 2D models. Additionally, improvements in the efficiency of point cloud processing and more advanced 3D positional embed-

dings could enhance the scalability and performance of these models across diverse 3D tasks, from autonomous navigation to complex scene understanding.

### D.3 POST-TRAINING PERFERENCE ALIGNMENT AND OPTIMIZATION

Preference alignment and optimization strategies in LLMs and MLLMs have become critical areas of research, particularly in addressing issues like hallucination (the generation of incorrect or false information) and the ethical implications of generating harmful or malicious content. Recent studies have contributed to a wide range of methodologies aimed at improving the alignment of model outputs with human expectations and ethical standards. These alignment strategies have been informed by the need to ensure models produce safe, coherent, and factually accurate outputs, while also avoiding ethical pitfalls, such as bias or harmful content generation (Huang et al., 2023a; Jiao et al., 2024).

Among the most commonly employed optimization techniques are those based on reinforcement learning with human feedback (RLHF). RLHF leverages human-provided labels to train models in a way that aligns their outputs with human preferences. The proximal policy optimization (PPO) algorithm (Schulman et al., 2017), a policy gradient method, is widely used in RLHF. It optimizes the model's policy by maximizing a reward function that reflects human preferences. However, while PPO and similar methods have demonstrated impressive performance, they suffer from significant computational overhead and sample inefficiencies. This is because policy gradient methods require a large number of samples and iterations to converge to optimal solutions, which leads to high resource consumption in large-scale models.

To address the limitations of RLHF and policy gradient approaches, a new class of optimization strategies has emerged. One prominent approach is Direct Preference Optimization (DPO) (Rafailov et al., 2024), which simplifies the optimization process by eliminating the need for complex policy gradient updates. Instead of training on policy rollouts, DPO focuses on a direct optimization objective based on pairwise comparisons of candidate outputs. Specifically, DPO operates in an offline manner, using preference pairs collected from human annotators to rank candidate outputs. By focusing on these pairwise preferences, DPO avoids the computational complexity of online training and the inefficiencies associated with traditional policy gradient methods. The model is trained to prefer outputs that rank higher in these pairwise comparisons, which leads to a more efficient alignment of the model's policy with human preferences.

Building on the foundation laid by DPO, several modifications have been proposed to further refine and enhance the method. One such adaptation is Identity-PO (Azar et al., 2024), which introduces identity mapping into the optimization process. Identity-PO focuses on directly optimizing pairwise preferences without relying on complex ranking mechanisms like ELO scores, which are often used in DPO. ELO-based ranking systems can lead to overfitting, where the model becomes overly specialized to the ranking system rather than generalizing well to new tasks. By using identity mapping, Identity-PO removes this reliance, leading to a more robust model that is less prone to overfitting and can generalize better across different types of tasks.

Another refinement is R-DPO (Park et al., 2024), which addresses a common issue in preference-based optimization: verbosity. Models trained on preference pairs often exhibit a tendency to generate overly verbose outputs, as longer outputs are frequently perceived as more informative and are thus preferred in the pairwise comparisons. To counter this issue, R-DPO introduces a length regularization term into the optimization process. This term discourages the model from generating excessively long outputs by penalizing verbosity, leading to more concise and relevant outputs. The regularization helps balance the trade-off between informative content and brevity, making the model's outputs more suitable for practical applications where verbosity can be problematic.

SimPO (Meng et al., 2024) further innovates on preference optimization strategies by eliminating the reference model, which is typically used as a baseline for comparing model outputs in many RLHF-based approaches. In SimPO, instead of comparing outputs to a fixed reference model, the optimization is based on the average log-likelihood of the model's outputs as an implicit reward. This approach simplifies the architecture by removing the dependency on a separate reference model, reducing the computational complexity and the risk of overfitting to a specific baseline. Additionally, using the average log-likelihood as a reward ensures that the model maintains a high degree of flexibility and generalization, as it is not tied to a specific reference.

## D.4 SELF-SUPERVISED LEARNING IN 3D UNDERSTANDING

Self-supervised learning methods have become increasingly prominent in the 3D domain, particularly for tasks involving complex geometric data. By enabling models to learn feature representations from unlabeled data, self-supervised learning reduces the need for large amounts of annotated data and has demonstrated significant potential in various 3D applications. Below are some key works and advancements in applying self-supervised methods to the 3D field.

One of the pioneering works in this area is PointContrast (Xie et al., 2020), which focuses on self-supervised learning for point cloud data. This method introduces a contrastive learning framework where the model learns discriminative features by contrasting different views of the same point cloud as positive samples and point clouds from different scenes as negative samples. By doing so, PointContrast enables the extraction of robust 3D point cloud representations, showing promising results in tasks like 3D point cloud matching and scene reconstruction.

Another significant contribution is STRL (Huang et al., 2021), which aims to learn dynamic representations of 3D objects from spatio-temporal data. STRL leverages 3D video data to capture both the geometric features of individual frames and the temporal motion of objects. This method has been successful in 3D action recognition and object tracking tasks, highlighting the effectiveness of self-supervised learning in dynamic 3D environments.

DepthContrast (Chhipa et al., 2022) focuses on self-supervised learning for depth images by utilizing the geometric structure information present in depth maps to learn 3D scene representations. Depth-Contrast treats depth maps as sparse representations of 3D scenes and uses a contrastive learning framework to align depth images from the same scene in a shared feature space while distinguishing depth maps from different scenes. This approach has demonstrated strong performance in scene understanding and 3D object detection tasks, showcasing the potential of self-supervised methods to extract meaningful 3D geometric information from depth images.

Another notable work in the 3D self-supervised learning space is OcCo (Wang et al., 2021), which designs a pretext task of completing occluded point clouds. The model is tasked with reconstructing complete 3D structures from partially observed point clouds, encouraging it to learn both global and local geometric features. OcCo's self-supervised pre-training significantly improves performance across downstream tasks such as 3D classification, semantic segmentation, and object detection, highlighting the efficacy of learning from occlusion-based tasks.

Contrastive Scene Contexts (Hou et al., 2021) introduces a novel self-supervised framework focused on learning spatial relationships between objects within a 3D scene. By leveraging the contextual information in 3D scenes, this method captures both semantic and geometric relationships. It uses contrastive learning by treating object pairs within the same scene as positive examples and object pairs from different scenes as negative examples, encouraging the model to learn discriminative spatial context features. This method has been successful in improving performance on 3D scene understanding and object retrieval tasks.

