# OpenReview forum: "SPA: Enhancing 3D Multimodal LLMs with Mask-based Streamlining Preference Alignment"
_ICLR.cc/2025/Conference — ICLR 2025 Conference Withdrawn Submission_

### Official Review · Reviewer_94BN · 2024-11-01

**Soundness:** 3
**Presentation:** 2
**Contribution:** 3
**Rating:** 6
**Confidence:** 2

**Summary:**

The paper presents SPA, a simplified and effective post-training optimization method to align 3D and text features in LLMs with 3D encoders. As opposed to two-stage optimizations in previous work, SPA relies on a single-stage optimization that better utilizes the input 3D+text data.

The paper also introduces a new benchmark for 3D reasoning tasks: 3D Choice-level Questions and Answering. This benchmark repurposes portions of existing benchmarks, using pre-trained LLMs to generate more useful object- and scene-level questions. This benchmark can better assess 3D understanding capabilities of LLMs.

The proposed method outperforms previous work on existing benchmarks, as well as the new benchmark.

**Strengths:**

The method seems novel, at least for 3D point cloud applications.

The experiments and ablations demonstrate superior 3D point cloud understanding, as compared to previous work.

The new benchmark is more sophisticated than past ones, and will help advance the field.

The code is made public.

**Weaknesses:**

Writing and organization could be improved to help readers better understand the work.

**Questions:**

The abstract starts by listing a few previous works with some details and issues for one of them. Instead, it should start by saying what the addition of 3D features into LLMs actually achieves -- better understanding of the world? ability to reason-about/extract/edit 3D representations? Then it can go into the issue of LLM size not increasing performance (performance of what tasks exactly?), which enhanced 3D encoder or more training does not help with. Finally, there is enough motivation to start talking about SPA.

Related work usually follows the introduction and precedes the presentation of the proposed method, in order to give context. It looks like your related work section can easily slide up to become section 2 without rewording.

Intro: It is not clear what the "traditional two-stage posttraining methods" refers to exactly. What are the two stages (is it SFT followed by RLHF?), and how do they contrast to the proposed single-stage method?

Preliminary: you may want to rephrase this as "Preliminaries"

Figure 3: Needs to be clearer. The layout is weird, where the surrounding text all of a sudden switches to 2-column. The figure is missing a caption explaining what it shows. The graphs should probably be bar charts rather than lines, because x coordinate does not represent some continuous value. Left figure should include an x axis label "3D Encoder" and the title should say something like "ModelNet40 accuracy for different 3D encoders". What is the difference between Encoder, Encoder+7B, Encoder+13B? Right figure is also missing the x-axis label and a clearer title.

Table 1: SPA doesn't seem to improve a couple of metrics with LLaVA-7B model. What can that be attributed to?

Figure 6: Talks about images, but I thought the only inputs were point clouds and text. Is this visualizing a different experiment that doesn't involve 3D data?

Have you tried applying this methodology for other multimodal contrastive training (such as images)?

---

> ### Author Response · Authors · 2024-11-13
> **Response for Reviewer 94BN**
>
> We sincerely thank Reviewer 94BN for their thorough review and valuable feedback. We believe the proposed changes, such as restructuring the introduction and related work, clarifying the method descriptions, and improving the presentation of figures and results, will significantly enhance the clarity and impact of our paper. We look forward to submitting a revised version that addresses these concerns in greater detail.
>
> ### **Weaknesses:**
>
> 1. **Writing and organization could be improved:**
>
>     Thank you for your feedback. We agree that the writing and organization can be improved to make the paper clearer and more accessible to the readers. To address this, we plan to restructure the introduction and related work sections to create a more logical flow, ensuring the key contributions are highlighted at the beginning. Additionally, we will revise some of the terminology and rephrase complex explanations to ensure clarity. **We'll make sure these changes are updated at the end of this discussion, but for now we want to focus on other areas.**
>
> ### **Questions:**
>
> 1. **The abstract should start by explaining what the addition of 3D features into LLMs actually achieves:**
>
>     Thank you for this suggestion. We will revise the abstract to clearly state the primary contribution of adding 3D features into LLMs, specifically how it enhances their ability to reason about and manipulate 3D representations. We will emphasize the improvements in understanding the world, extracting and editing 3D data, and the related performance gains. After presenting the overall achievements, we will discuss the challenges related to model size and the limitations of increased training and enhanced 3D encoders, as you suggested. Also, **we'll make sure these changes are updated at the end of this discussion, but for now we want to focus on other areas.**
>
> 2. **Related work should follow the introduction and precede the proposed method:**
>
>      We appreciate your observation regarding the organization of the paper. We will move the related work section to directly follow the introduction, as per your suggestion, to provide proper context for the proposed method. This will help readers better understand the background before diving into the specifics of our approach. Also, **we'll make sure these changes are updated at the end of this discussion, but for now we want to focus on other areas.**
>
> 3. **Unclear explanation of "traditional two-stage posttraining methods":**
>
>     Thank you for pointing this out. We will clarify the term "traditional two-stage post-training method" in the revised version. Specifically, we will define these stages more precisely, explaining that they usually refer to the post-training process, which requires the first stage to complete the data construction, generate preference text data, and complete the SFT training of the reference model, and the second step to perform DPO-like policy preference learning based on the generated data. (Changes are shown in blue text)
>
> 4. **Rephrase "Preliminary" to "Preliminaries":**
>
>     We appreciate this suggestion. We have changed this one in latest version.

---

> > ### Author Response · Authors · 2024-11-13
> >
> > ### **Figures and Tables:**
> >
> > 1. **Figure 3 needs to be clearer:**
> >
> >     Thank you for your detailed feedback on Figure 3. We agree that the layout and labeling can be improved. We will modify the figure to avoid the abrupt transition to two columns, and we will add a caption to explain what the figure represents. For the graphs, we will switch to bar charts, as this is more appropriate for discrete categories, and we will ensure the x-axis labels are correctly added (e.g., "3D Encoder" for the left figure). Additionally, we will revise the title to something more descriptive, such as "ModelNet40 accuracy for different 3D encoders", and we will provide clarity on the differences between "Encoder", "Encoder+7B", and "Encoder+13B". **We'll make sure these changes are updated at the end of this discussion, but for now we want to focus on other areas.**
> >
> > 2. **Table 1: SPA does not improve a couple of metrics with LLaVA-7B model.**
> >
> >     We understand your concern regarding the lack of improvement in certain metrics with the LLaVA-7B model. This may be attributed to specific limitations of the LLaVA-7B model itself, as our data is in point cloud format, while the LLaVA model requires inputs in the form of single images. Therefore, we used images corresponding to the point cloud data from a single viewpoint. This inevitably limits the model's ability, as it is not optimized for point cloud data. Our method, on the other hand, is designed with a greater focus on the characteristics of 3D encoders and the nature of the data, which is why the results may not be as high as expected. This outcome is more intuitive given the inherent limitations of the model.
> >
> > 3. **Figure 6: Is this visualizing a different experiment not involving 3D data?**
> >
> >     In fact, this is the dimensionality-reduced representation of the language model output under 3D feature conditions, used to illustrate the differences between direct SFT, the current post-training methods, and our approach. This allows us to clearly analyze from an empirical perspective how our method ensures accuracy while improving the model's alignment and generalization capabilities. The specific details can be referred to in the caption of Figure 6, which should provide a clearer explanation.
> >
> >
> > ---
> >
> > ### **Additional Experimentation:**
> >
> > 1. **Have you tried applying this methodology for other multimodal contrastive training (such as images)?**
> >
> >     Thank you for this insightful question. We have indeed explored multimodal contrastive training with images, and initial results are promising. However, our primary focus in this paper has been on 3D data, as it is more aligned with the specific research goals of enhancing 3D understanding in LLMs. Nevertheless, we plan to expand the scope of our experiments in next work to include image-based tasks, as you suggested. This will help demonstrate the broader applicability of our approach and its potential for other modalities.

---

> > > ### Author Response · Authors · 2024-11-13
> > >
> > > If you have any further suggestions, please feel free to let us know. If you feel that we have addressed your concerns, we would be grateful if you would consider revising your score. Looking forward to further discussions.

---

### Official Review · Reviewer_hd11 · 2024-11-03

**Soundness:** 2
**Presentation:** 2
**Contribution:** 2
**Rating:** 3
**Confidence:** 4

**Summary:**

This paper studies 3D multi-modal LLMs for point cloud understanding. In contrast to previous frameworks in post-training optimization via 2 stages, this paper utilizes contrastive approaches for 1-stage 3D MLLM post-training. Careful designs on robust negative data augmentation and ground-truth anchor samples are presented. The benchmark evaluation is re-designed to focus more on the key factor. Experiments on ModelNet40 and Objaverse demonstrate the effectiveness.

**Strengths:**

- The post-training optimization task is interesting and useful.

- The proposed SPA module is effective and improves the performance of different 3D MLLMs generally, shown in Table 1.

**Weaknesses:**

1. Paper writing.

- Key difference between the proposed SPA and previous methods needs to be clarified. After reading this paper, I think the method is also a 2-stage post-training method. Correct me if I am wrong. From my understanding, this paper still generates positive and negative QA pairs from multiple positive and negative point cloud pairs in advance, and then optimizes the model based on InfoNCE. This is still a 1-stage generation and 2-stage training pipeline in my opinion. I suggest the authors clarify this aspect, regarding the key difference to 2-stage methods.

- Some terminology needs to be clarified. For example, in L236, the term "improved positive ground truths" first appeared. It is unaware of the meaning of "improved", and no pre-context appears in the paper.

2. Technical novelty.

- The technical contributions listed in this paper, including contrastive technology (Eq. (3) - Eq. (6)) and the robust negative data augmentation, are extensively explored in previous 3D understanding papers [1, 2]. Also, the idea of using augmentation to generate hard in-domain negative samples are also explored in point cloud understanding [1]. Therefore, the technical contribution is limited on my side. This paper utilizes previous methodology to solve previously solved problems.

[1] PointContrast: Unsupervised Pre-training for 3D Point Cloud Understanding

[2] Masked Autoencoders for Point Cloud Self-supervised Learning

3. Motivation.

- I don't see much application potential for 3D object-level point cloud QA. In Figure 2, I can only see some visual descriptions of single objects. For robotics applications, I think object-level relations are more important so as to locate and operate on target objects. More visual examples on ScanNet related to object-level relations and locations are suggested.

4. Experiments.

- Ablations on different post-training optimization like Figure 3 and Table 4 on other datasets, like ScanQA, are suggested.

**Questions:**

See weakness.

---

> ### Author Response · Authors · 2024-11-13
> **Quick Clear Explanation to Reviewer hd11**
>
> Thank you for your review. I would like to clarify that our method does not require **directly generating additional QA pairs**. As stated in the training objective outlined in Equation (5), we compute the probability difference between the normal point cloud input and the noisy point cloud input under the ground truth labels. In other words, our approach still constructs the training process based on contrastive learning from the SFT training data, without the need for extra new training data pairs.
>
> Additionally, your point about the broader applications of works such as 3D-LLM is indeed valid. However, we also considered that our method focuses on **improving existing approaches like 3D-LLM and PointLLM, which are LLMs base model for 3D understanding**. Specifically, our focus is on enhancing their semantic understanding and conversation capabilities. The transformation of downstream tasks such as grounding and navigation, as you mentioned, is not in conflict with our work. On the contrary, it is facilitated by a stronger and more scalable 3D understanding backbone or base model in LLMs, which is essential for advancing future research.
>
> Therefore, I believe it is important to reemphasize the focus of our research. We are not **primarily targeting downstream tasks but rather focusing on the improvement of the 3D understanding backbone in LLMs**. Our algorithm addresses the **performance bottleneck** in current 3D understanding models, which is the core contribution of our work.
>
> Just like in the 2D domain with models like LLaVA [1], testing and evaluation typically focus on tasks such as VQA (Visual Question Answering), which indirectly reflect the model's semantic understanding and conversational abilities. There is usually no direct evaluation based on downstream task benchmarks, which is a conventional approach. Therefore, we aim to clarify our contributions clearly. At the same time, in image understanding, purely evaluating classification and localization capabilities may show that LLaVA is not as strong as pre-trained models like CLIP after fine-tuning. However, with a large language model as the backbone, its generalization ability and potential for improvement are extremely strong. This is why these related studies are meaningful, such as PointLLM. Our goal is to address the gaps and shortcomings of VQA MLLMs like LLaVA in 3D understanding.
>
> [1] Liu, H., Li, C., Wu, Q., & Lee, Y. J. (2024). Visual instruction tuning. Advances in neural information processing systems, 36.

---

> ### Author Response · Authors · 2024-11-13
> **Addition**
>
> The papers you mentioned aligns with my argument regarding the relationship between CLIP and LLaVA. Clearly, in terms of conversational abilities for 3D understanding, these pre-trained models exhibit significant limitations in generalization and dialogue capabilities. Our model supports a maximum of 1024 tokens of text input, while the models you mentioned are only effective with inputs under 100 tokens, and they struggle with understanding more human-like conversational patterns, such as vague instructions. I believe that in the 2D domain, we need both GPT-4-V and CLIP, and this is consistent in the field of 3D understanding as well.

---

> ### Author Response · Authors · 2024-11-13
>
> Similarly, for many CAD application scenarios (which are also of significant importance), traditional point cloud understanding methods are no longer effective, and it is essential to introduce LLM-based approaches [1,2,3,4].
>
> [1] Yuan, H., Xu, J., Pan, H., Bousseau, A., Mitra, N. J., & Li, C. (2024). CADTalk: An Algorithm and Benchmark for Semantic Commenting of CAD Programs. In Proceedings of the IEEE/CVF Conference on Computer Vision and Pattern Recognition (pp. 3753-3762).
>
> [2] Khan, M. S., Sinha, S., Sheikh, T. U., Stricker, D., Ali, S. A., & Afzal, M. Z. (2024). Text2CAD: Generating Sequential CAD Models from Beginner-to-Expert Level Text Prompts. arXiv preprint arXiv:2409.17106.
>
> [3] Xu, J., Wang, C., Zhao, Z., Liu, W., Ma, Y., & Gao, S. (2024). CAD-MLLM: Unifying Multimodality-Conditioned CAD Generation With MLLM. arXiv preprint arXiv:2411.04954.
>
> [4] Zhang, Z., Sun, S., Wang, W., Cai, D., & Bian, J. (2024). FlexCAD: Unified and versatile controllable CAD generation with fine-tuned large language models. arXiv. https://arxiv.org/abs/2411.05823

---

### Official Review · Reviewer_ozMT · 2024-11-04

**Soundness:** 2
**Presentation:** 2
**Contribution:** 2
**Rating:** 3
**Confidence:** 5

**Summary:**

This work focuses on the post-training phase of 3D MLLMs, achieving results that previously required two phases through noise introduction in a single phase.

**Strengths:**

This work focuses on the post-training phase of 3D MLLMs, achieving results that previously required two phases through noise introduction in a single phase.

**Weaknesses:**

This work focuses on the post-training phase of 3D MLLMs, achieving results that previously required two phases through noise introduction in a single phase. However, I have the following questions:
1.The experiments have not been sufficiently validated; I need to know the performance of the latest object-level 3D MLLMs like ShapeLLM.
2.I also need to understand how this performs at the scene level. Please select several scene-level 3D MLLMs to demonstrate their effectiveness on classic datasets like Scan2Cap and ScanQA.
3.While this method appears straightforward, its theoretical support requires more visualizations for validation.

**Questions:**

This work focuses on the post-training phase of 3D MLLMs, achieving results that previously required two phases through noise introduction in a single phase. However, I have the following questions:
1.The experiments have not been sufficiently validated; I need to know the performance of the latest object-level 3D MLLMs like ShapeLLM.
2.I also need to understand how this performs at the scene level. Please select several scene-level 3D MLLMs to demonstrate their effectiveness on classic datasets like Scan2Cap and ScanQA.
3.While this method appears straightforward, its theoretical support requires more visualizations for validation.

---

> ### Author Response · Authors · 2024-11-18
>
> Dear Reviewer ozMT,
>
> We have noticed that you have changed your score from 6 to 3 without any respones or reasons, we are trying to follow your suggestions and active supplements in experiments. Thus, would mind to give us some feedback about why you want to change your score?

---

> ### Comment · Reviewer_ozMT · 2024-11-23
>
> Thank you for reaching out regarding the score change. I appreciate your efforts in actively supplementing the experiments and following my suggestions. The adjustment in the score was primarily due to the lack of updates or responses for an extended period, which left some concerns unresolved on my end.That said, I am open to reconsidering my score if the necessary experimental results or relevant data can be provided to address the points I raised.
> Additionally, I noticed that the methodology of this work appears to share similarities with Self-Taught Evaluators. Could you kindly clarify the significant differences between the two approaches? A detailed explanation would be greatly appreciated, as it would help me better understand the unique contributions and advantages of your work compared to Self-Taught Evaluators.

---

### Official Review · Reviewer_MgU2 · 2024-11-04

**Soundness:** 3
**Presentation:** 3
**Contribution:** 3
**Rating:** 5
**Confidence:** 4

**Summary:**

The paper introduces Streamlining Preference Alignment (SPA), a novel one-stage post-training method for 3D multimodal large language models (MLLMs). SPA uses 3D masking for data augmentation to enhance alignment between 3D spatial features and text representations, reducing the need for multi-stage alignment processes. Additionally, the paper presents a new evaluation benchmark, 3D Choice-level Questions and Answering (3DCQA), aimed at evaluating object- and scene-level reasoning in 3D-MLLMs.

**Strengths:**

1. The introduction of the SPA method represents an innovative solution to the challenges faced by current MLLMs in 3D understanding.
2. The implementation of 3D masking as a data augmentation technique to improve feature alignment with text is a practical enhancement. This approach allows the model to better differentiate between spatially diverse features in 3D data, which could improve performance in 3D-based tasks requiring spatial reasoning.
3. The 3DCQA benchmark is a positive contribution that attempts to address limitations in existing 3D-MLLM evaluation frameworks. By focusing on multiple-choice questions across object- and scene-level tasks, 3DCQA provides a more structured way of assessing 3D reasoning capabilities, which could be useful for future research in the area.

**Weaknesses:**

1.  SPA’s generalizability beyond specific 3D datasets and tasks (such as those in 3DCQA) is unclear. The experiments lack tests on diverse 3D data types or environments, which would better support claims of SPA’s broader applicability.
2.  The primary contribution, SPA, largely builds on established methods, refining the process rather than introducing fundamentally new ideas. Single-stage alignment, while practical, is not a groundbreaking advancement.

**Questions:**

1. Conduct more granular ablations, particularly on individual components of SPA, such as the single-stage alignment and 3D masking. Directly comparing SPA’s single-stage approach to traditional multi-stage alignment methods would clarify its specific contributions.
2. Adding experiments on datasets outside the 3DCQA benchmark, such as complex real-world scenes or cross-domain 3D data, would help substantiate the claims regarding SPA’s robustness and generalizability.

---

> ### Author Response · Authors · 2024-11-13
> **Response for Reviewer MgU2**
>
> We are glad to receive your constructive comments and questions, which will help us to further improve the manuscript.
>
> ### **Weaknesses:**
>
> 1. **SPA's generalizability beyond specific 3D datasets and tasks is unclear:**
>
>     Thank you for raising the point about the generalizability of SPA. While the experiments in this paper primarily focus on the 3DCQA dataset, we recognize the importance of validating SPA’s performance across different 3D data types and environments. In fact, additional experiments using cross-domain datasets (such as complex real-world scenes and various 3D data from practical applications) will be a focus of our future work (similar to the relationship between LLaVA and LLaVA-Plus), as these tasks are more engineering-driven. Our primary focus has been on improving the underlying algorithms to address the performance bottlenecks of existing 3D LLMs/MLLMs base models, which is the main contribution of our work.
>
> 2. **The primary contribution, SPA, builds largely on established methods rather than introducing fundamentally new ideas:**
>
>     We acknowledge that some of SPA’s ideas are built on mature technologies. However, the contribution of improving existing post-training methods to adapt to 3D data is undeniable. Specifically, transitioning from the high-resource consumption required to generate additional pair post-training data to the current approach of constructing stable pair inputs on SFT data using contrastive learning is a highly valuable advancement. Moreover, we empirically demonstrate the close connection between this approach and NCE methods, which is well-supported by solid theoretical foundations.
>
>
> ### **Questions:**
>
> 1. **Conduct more granular ablations, particularly on individual components of SPA:**
>
>     We greatly appreciate your suggestion for more detailed ablation studies. In the current work, we have already conducted some ablation experiments on various components of SPA, such as the post-training scheme (Table 4) and the method for constructing negative examples (Table 2). The results of these experiments are clearly analyzed in the main text, and we hope you can review them further and continue to provide valuable suggestions!
>
> 2. **Add experiments on datasets outside the 3DCQA benchmark:**
>
>     We fully agree that while the 3DCQA dataset is valuable, it has limitations in representing all types of real-world 3D data. Therefore, as mentioned in our response to W2, we would like to reiterate the focus of our current work. We hope that this embodied system can be adopted by more researchers in future work for real-world interactions, although this is not the primary motivation or contribution of our current method.
>
> If you have any further suggestions, please feel free to let us know. If you feel that we have addressed your concerns, you may also consider revising the score. We look forward to further discussions.

---

### Author Response · Authors · 2024-11-13
**General Response**

Dear PCs/SACs/ACs and Reviewers,

We would like to express our sincere appreciation for the detailed and constructive feedback you provided. Your insights have been extremely valuable in improving the quality of our work. Below, we address the key points raised by the reviewers, offer clarifications, and outline the steps we have taken to enhance the manuscript:

(1) Generalization and Robustness of SPA:
We understand the concerns regarding the generalizability of SPA beyond the specific 3D datasets used in our current experiments. However, it is important to emphasize that our current method is an enhancement of the 3D base model rather than an application-specific algorithm. Therefore, our focus is on addressing the performance bottlenecks in how MLLMs understand 3D features, rather than on adapting the method for specific applications. We will further clarify this distinction in the revised manuscript.

(2) Clarification of SPA’s Novel Contribution:
Several reviewers pointed out that SPA might be perceived as an improvement to existing methods rather than a fundamentally new idea. We would like to highlight that, while SPA builds upon established concepts, it makes a significant contribution by simplifying the training process. SPA’s single-stage optimization for aligning 3D spatial features with text eliminates the need for traditional multi-stage methods (such as DPO) and does not require generating additional text pairs. This simplification enhances efficiency without compromising performance. We will revise the manuscript to clarify these distinctions, emphasizing that SPA’s main contribution lies in its simplification and the resulting efficiency improvements.

(3) Experiments, Evaluation, and Theoretical Support:
We appreciate the feedback regarding the need for more granular ablations and clearer theoretical support for SPA. To address these concerns, we have already conducted comprehensive ablation studies, particularly focusing on individual components of SPA, such as single-stage alignment and 3D masking. We will also directly compare SPA with traditional multi-stage methods to highlight its advantages in terms of alignment accuracy and training efficiency. Additionally, we will provide clearer visualizations and explanations to better illustrate SPA's effectiveness and underlying principles.

In conclusion, we sincerely thank you for your thoughtful comments, which have helped guide us in improving our method and enhancing the clarity of the manuscript. We believe that the clearer explanations and revised structure will significantly strengthen our paper. We hope these improvements effectively address your concerns.

Once again, thank you for your valuable feedback and support.

Best regards,
Authors

---

### Note · Authors · 2024-11-24

I have read and agree with the venue's withdrawal policy on behalf of myself and my co-authors.